# Epistatic interaction between *ERAP2* and HLA modulates HIV-1 adaptation and disease outcome in an Australian population

Marwah Al-kaabi[1], Pooja Deshpande[1,2¤a], Martin Firth[3], Rebecca Pavlos[2], Abha Chopra[2], Hamed Basiri[1], Jennifer Currenti[1¤b], Eric Alves[1], Spyros Kalams[4], Jacques Fellay[5,6], Elizabeth Phillips[2,4], Simon Mallal[2,4], Mina John[1,2,7], Silvana Gaudieri[1,2,4]*

1 School of Human Sciences, University of Western Australia, Crawley, Australia, 2 Institute for Immunology and Infectious Diseases, Murdoch University, Murdoch, Australia, 3 School of Physics, Mathematics and Computing, Department of Mathematics and Statistics, University of Western Australia, Crawley, Australia, 4 Division of Infectious Diseases, Department of Medicine, Vanderbilt University Medical Center, Nashville, Tennessee, United States of America, 5 School of Life Sciences, École Polytechnique Fédérale de Lausanne, Lausanne, Switzerland, 6 Swiss HIV Cohort Study, Zurich, Switzerland, 7 Department of Clinical Immunology, Royal Perth Hospital, Perth, Australia

¤a Current address: Perron Institute for Neurological and Translational Science, Nedlands, Australia
¤b Current address: Curtin Medical School, Curtin University, Bentley, Australia
* silvana.gaudieri@uwa.edu.au

**Data Availability Statement:** GenBank accession's numbers of HIV sequences and individual host

## Abstract

A strong genetic predictor of outcome following untreated HIV-1 infection is the carriage of specific alleles of human leukocyte antigens (HLAs) that present viral epitopes to T cells. Residual variation in outcome measures may be attributed, in part, to viral adaptation to HLA-restricted T cell responses. Variants of the endoplasmic reticulum aminopeptidases (ERAPs) influence the repertoire of T cell epitopes presented by HLA alleles as they trim pathogen-derived peptide precursors to optimal lengths for antigen presentation, along with other functions unrelated to antigen presentation. We investigated whether ERAP variants influence HLA-associated HIV-1 adaptation with demonstrable effects on overall HIV-1 disease outcome. Utilizing host and viral data of 249 West Australian individuals with HIV-1 subtype B infection, we identified a novel association between two linked *ERAP2* single nucleotide polymorphisms (SNPs; rs2248374 and rs2549782) with plasma HIV RNA concentration (viral load) (P adjusted = 0.0024 for both SNPs). Greater HLA-associated HIV-1 adaptation in the HIV-1 *Gag* gene correlated significantly with higher viral load, lower CD4+ T cell count and proportion; P = 0.0103, P = 0.0061, P = 0.0061, respectively). When considered together, there was a significant interaction between the two *ERAP2* SNPs and HLA-associated HIV-1 adaptation on viral load (P = 0.0111). In a comprehensive multivariate model, addition of *ERAP2* haplotypes and HLA associated adaptation as an interaction term to known HLA and CCR5 determinants and demographic factors, increased the explanatory variance of population viral load from 17.67% to 45.1% in this dataset. These effects were not replicated in publicly available datasets with comparably sized cohorts, suggesting that any true global epistasis may be dependent on specific HLA-ERAP allelic combinations. Our data raises the possibility that *ERAP2* variants may shape peptide repertoires presented to HLA class I-restricted T cells to modulate the degree of viral adaptation within

genetic data are provided in 'S1 Data and S2 Data' supplementary information files.

**Funding:** This work was funded by the National Health and Medical Research Council (APP1148284) grant awarded to SG, MJ, SK and SM; National Institutes of Health-funded Tennessee Center for AIDS Research grant (P30AI110527) awarded to SM and SK; University of Western Australia grant (2022/GR000787) awarded to SG; Government of Western Australia grant (2021/GR000223) awarded to SG; BioZone Collaboration Grant awarded to EA; BioZone Completion Scholarship awarded to EA; Australian Government Research Training Program awarded to EA; Australian Government Research Training Program Fees Offset awarded to MA; The Swiss HIV Cohort Study (201369) awarded to JF. The funders had no role in study design, data collection and analysis, decision to publish, or preparation of the manuscript.

**Competing interests:** The authors have declared that no competing interests exist.

individuals, in turn contributing to disease variability at the population level. Analyses of other populations and experimental studies, ideally with locally derived ERAP genotyping and HLA-specific viral adaptations are needed to elucidate this further.

## Author summary

HIV infection outcome is variable between individuals and understanding the factors that impact this variation is important for efforts towards a HIV cure or vaccine. Here, we found that the level of HIV in the blood is affected by whether an individual carries a specific form of *ERAP2*, a molecule that influences processing and presentation of the virus to the immune system, as well as the degree to which HIV has mutated to adapt to immune responses. We also show that the interaction of *ERAP2* and other known genetic factors explains greater variation in infection outcome than these factors alone. These findings expand our knowledge of the potential importance of viral processing and presentation in the immune response to HIV.

## Introduction

Human immunodeficiency virus-1 (HIV-1) infection continues to present a global health challenge with no existing curative therapy or effective vaccine. Understanding the impact of host and viral factors on the natural course of HIV-1 infection remains an ongoing area of investigation. Importantly, exploring the influence of genetic factor interactions, such as epistasis, rather than individual single nucleotide polymorphisms (SNPs) on HIV-1 infection outcome will allow a better understanding of disease pathogenesis [1].

A major host immune-related genetic factor that impacts plasma HIV RNA concentration (viral load) are the human leukocyte antigen (HLA) class I loci [2–4], which encode proteins that act as antigen presenting molecules to T cells and as NK cell receptor ligands. The HLA class I alleles B*57 and B*27 have been shown to be strong predictors of lower viral load and slower disease progression [5]. However, about 30% of individuals with HIV infection who maintain robust viral control do not carry these protective HLA-B alleles [6]. Furthermore, the combined variations in the HLA class I region and in CCR5 (32 bp deletion within the receptor for HIV-1 cell binding and entry) only explain ~25% of set-point viral load (spVL) in genome wide association studies (GWAS), where set-point refers to a point of equilibrium following the peak in viral load after acute infection [4]. This highlights the need to identify additional factors associated with HIV-1 infection outcome and/or factors that modulate the effect of HLA alleles.

The endoplasmic reticulum aminopeptidases 1 (*ERAP1*) and 2 (*ERAP2*) are polymorphic genes that have been found to interact with HLA class I genes and influence clinical outcome in several cancers, autoimmune and infectious diseases [7–9]. These genes encode enzymes that trim N-terminal peptide precursors into an optimal length to accommodate the highly variable HLA binding groove [10,11]. Thereby, *ERAP1* and *ERAP2* define the peptide repertoire presented to T cells restricted by specific HLA genotypes [12]. As genetic variations in the ERAP loci mark a broad range of enzymatic activities that vary in terms of catalytic efficiency and substrate affinity [13], greater clarity is needed on how variation in these *ERAP* genes impact HLA-restricted T cell immune responses. While polymorphisms in the *ERAP1* and *ERAP2* genes are found to alter peptide repertoire [9,14,15], recent evidence also suggests

an influence of *ERAP2* on cytokine expression [16] and other factors involved in modulating anti-viral immune responses [17]. ERAP molecules may therefore affect HIV-1 infection outcome by mechanisms that are independent of peptide repertoire effects on antigen presentation.

Viral genetic diversity is another factor that contributes to HIV-1 infection outcome. The observed genetic diversity of HIV-1 includes specific polymorphisms (adaptations) selected by HLA-restricted T cell immune pressure [18], with the level of adaptation in the autologous virus associated with infection outcome [19,20]. The selection of de novo adaptations or reversion of transmitted viral adaptations during HIV-1 infection reflects the cost of the specific polymorphism to the replicative fitness of the virus in the selective immune environment of the host [20,21]. Shifts in the targeted peptide repertoire would be predicted to alter where adaptations occur and in turn, the overall balance between immune escape and replicative capacity cost as reflected by viral load.

In this study, we aimed to investigate viral adaptations and host genetic factors and their association with viral load and other disease markers in a well characterized cohort of people living with HIV-1 infection from Western Australia (WAHIV) [22]. The historical nature of this cohort enabled the examination of host and viral factors on pre-treatment viral load. Access to pre-treatment viral load data is increasingly challenging in cohort studies due to early or immediate initiation of antiretroviral therapy (ART) becoming standard of care [23]. We sought to investigate the potential role of *ERAP2* in altering HIV-1 infection outcome and if there was evidence of additive or synergistic interaction between ERAP and HLA.

## Results

### *ERAP2* significantly influences HIV-1 infection outcome

Our previous study of the WAHIV cohort has shown that carriage of the HLA class I alleles B*57:01 and B*27:05 are associated with lower pre-treatment viral load [24]. We extended our earlier study to determine whether variation in *ERAP1* and *ERAP2* influences infection outcome. Initially, we investigated the non-independent impact of two main *ERAP2* SNPs rs2549782 and rs2248374 [25] on HIV-1 infection outcome. Both SNPs encode for notable changes at the functional level of the enzyme: rs2549782 is a missense variant [25] that switches enzyme specificity and activity [26] and rs2248374 is an intron variant that disrupts RNA splicing leading to the loss of *ERAP2* expression [25]. The two SNPs are in strong linkage disequilibrium (LD; $r^2$ = 0.936, D'= 0.975; Fig 1A and 1B). Association analysis using different genetic inheritance models showed a strong significant association with viral load for both SNPs, in which the presence of the G allele of the rs2248374 SNP and the T allele of the rs2549782 SNP associate with lower viral load (P adjusted Codominant = 0.0081, P adjusted Dominant = 0.0024 for both SNPs; S1 Table). A trend with CD4$^+$ T cell count was identified for both SNPs (P adjusted Dominant = 0.05), while no association was detected with CD4$^+$ T cell proportion after false discovery rate (FDR) correction (S1 Table).

### HIV-1 infection outcome is not affected by *ERAP1* genotype

Previous studies suggest that variations in *ERAP1* can alter clinical outcome in certain infections [9]. We sought to investigate whether this effect can be detected in participants with HIV infection. We screened for an independent association signal for 10 *ERAP1* SNPs with HIV-1 infection outcome; eight of the SNPs are the most frequent *ERAP1* SNPs (with >1% frequency) reported [13,27] and shown to be a risk factor for many diseases [28,29]. Three *ERAP1* SNPs (rs26653, rs27895 and rs3734016) showed a significant association with viral load (rs26653: P Codominant = 0.0462, P Recessive = 0.0131; rs27895: P Codominant = 0.0411,

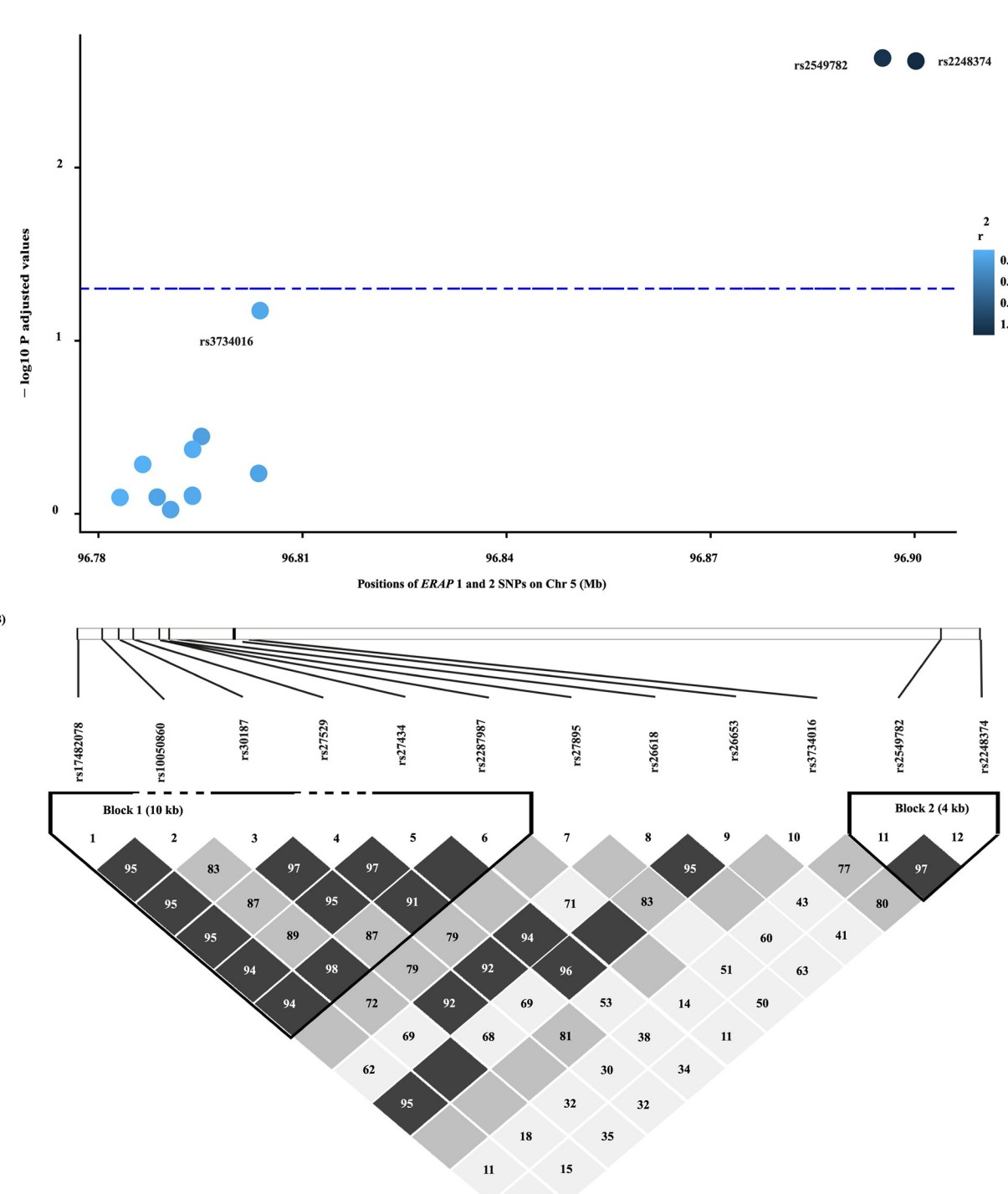

**Fig 1. *ERAP2* SNPs associated with viral load in HIV-1 infected participants. A)** Manhattan plot for *ERAP1* and *ERAP2* SNPs and association with HIV-1 disease outcome measured by log viral load (RNA copies/ml). Negative $\log_{10}$ FDR adjusted P values are plotted against the physical position of each SNP on chromosome 5. The blue dashed line represents the nominal significant level 0.05. The dominant model of inheritance is applied comparing the homozygous major alleles versus the heterozygous and the homozygote minor alleles together. The SNPs are coloured according to their linkage strength measured by correlation coefficient ($r^2$) with the tag SNP rs2248374. The ANOVA statistical analysis was

performed using the SNPassoc R package. Sex, race and CCR5Δ32 deletion were included in the statistical model as covariates. The analysis included 249 treatment naïve HIV-1 infected participants. The plot was generated in RStudio software. **B)** Linkage disequilibrium plot showing the linkage patterns of the *ERAP1* and *ERAP2* SNPs measured by Dʻ. The *ERAP1* and *ERAP2* SNPs are ordered according to their position in the human genome. Confidence bounds colour scheme is used to represents the strength of the LD patterns among *ERAP1* and 2 SNPs, where the dark grey colour reflects strong evidence of LD and the white reflects independent genotypes. The values of the Dʻare displayed in each square and the dark grey squares without numbers indicate Dʻequals to 1. Plot was constructed using HaploView 4.2 software [74].

P Recessive = 0.0172; rs3734016: P Codominant = 0.0483, P Dominant = 0.0163, P Overdominant = 0.0138). However, these associations did not reach significance following FDR correction (S1 Table; trend for rs3734016, P adjusted = 0.0652). When investigating the association with CD4$^+$ T cell count, rs26653 also had a significant association under the recessive model (P = 0.0392) and rs26618 under the dominant model (P = 0.0442) but did not reach significance following FDR correction. A similar association with CD4$^+$ T cell proportion was observed for rs26653 and rs3734016 (rs26653: P Codominant = 0.0279, P Recessive = 0.0076; rs3734016: P Dominant = 0.0399, P Overdominant = 0.0438) (S1 Table). Both SNPs are in strong LD with other SNPs in the *ERAP1* gene region (Fig 1A and 1B).

## *ERAP2* haplotype B, carrying a truncated version of *ERAP2*, predicts better HIV-1 infection outcome

ERAP haplotypes were constructed to investigate the additive effect of *ERAP1* and *ERAP2* variants on disease outcome. *ERAP2* haplotypes are defined based on the alteration in gene transcriptional length caused by the SNP rs2248374. Haplotype A is associated with a full-length *ERAP2* protein, whereas haplotype B is associated with a shortened protein that is subject to nonsense-mediated decay of the mRNA [25]. The majority of the cohort carry either haplotype A (rs2549782G and rs2248374A, 51.1%) or haplotype B (rs2549782 T and rs2248374 G, 47.4%), with a few participants carrying different combinations (i.e. rs2549782 G and rs2248374 G or rs2549782 T and rs2248374 A, 1.5%; S2 Table). The frequency of the SNP combinations is comparable to the frequencies observed by others [13].

Overall, *ERAP2* haplotypes associated significantly with differences in viral load (P = 0.0007, $\eta_p^2$ = 0.06, ANOVA; Fig 2A) and CD4$^+$ T cell count (P = 0.0088, $\eta_p^2$ = 0.04; Fig 2B) more so than CD4$^+$ T cell proportion (P = 0.097, $\eta_p^2$ = 0.02; Fig 2C) (S3 Table). In particular, carriage of haplotype B was associated with a lower viral load (P adjusted = 0.0002, $\eta_p^2$ = 0.06, ANOVA), higher CD4$^+$ T cell count (P adjusted = 0.0030, $\eta_p^2$ = 0.04) and higher CD4$^+$ T cell proportion (P adjusted = 0.0363, $\eta_p^2$ = 0.02) (S3 Table).

We utilized the genotype data of eight *ERAP1* SNPs to construct haplotypes for the study participants. The frequency of the estimated *ERAP1* haplotypes in the cohort reflected frequencies observed in different populations [13] (S4 Table). However, due to the high number of *ERAP1* haplotype patterns in this cohort, this study was underpowered to investigate the association of *ERAP1* haplotypes with disease outcome.

## Viral adaptations to HLA-restricted T cell immune responses in *Gag* correlate significantly with clinical outcome

Viral adaptations to HLA-restricted CD8$^+$ T cell immune responses have been shown to influence HIV-1 infection outcome [19,20]. Using each individual's autologous HIV-1 sequence and HLA repertoire, we found that high levels of viral adaptation in *Gag* to host HLA alleles correlated significantly with high viral load and lower CD4$^+$ T cell count and proportion (P = 0.0103, P = 0.0061, P = 0.0061, respectively; S5 Table and Fig 3). This remained significant when limiting the analysis to subtype B only (P = 0.0250, P = 0.0129, P = 0.0144, respectively), as the major

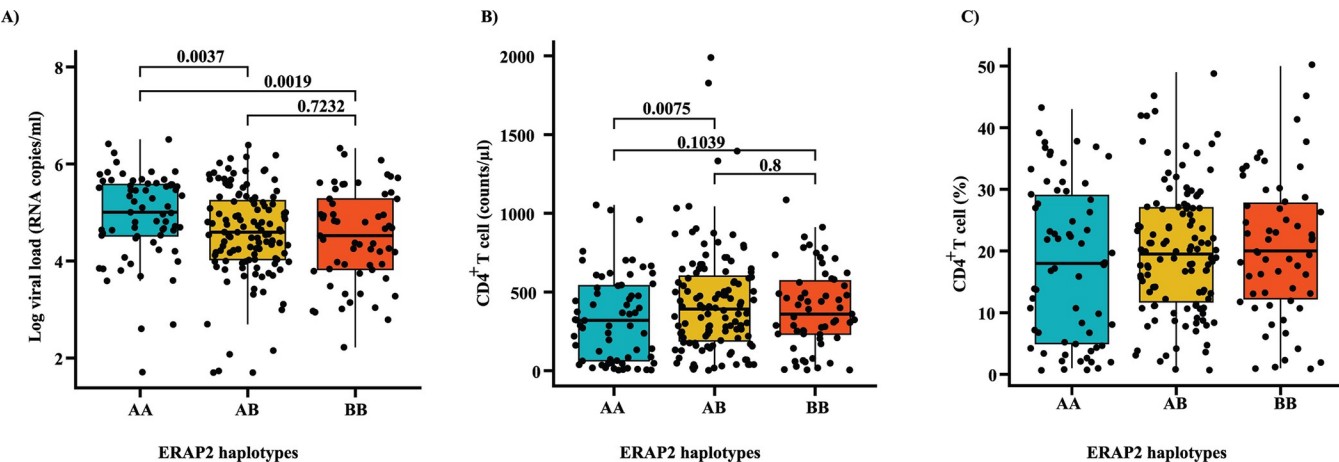

**Fig 2. Variation in outcome measures following HIV-1 infection is likely influenced by *ERAP2* haplotypes. A)** Participants carrying Haplotype B, the truncated type of *ERAP2*, have lower viral load than participants with Haplotype A. **B)** High CD4+ T cell counts is associated with Haplotype B. **C)** No significant difference in CD4+ T cell proportion in participants with different *ERAP2* haplotype combinations. Analysis performed using ANOVA model adjusted for sex, race and CCR5Δ32 deletion.

HIV subtype in the cohort (Table 1). In contrast, *Pol* and *Nef* adaptation scores did not show a correlation with the clinical markers, even when restricted to subtype B. (S5 Table).

Examining the level of autologous viral adaptation scores across the HIV-1 genome did not show significant differences between the genes examined in this study (P = 0.162, Kruskal-Wallis test; S1 Fig).

## Interaction effect of *ERAP2* with viral adaptation may shape outcome following HIV-1 infection

We further sought to examine the interaction effect of *ERAP2* haplotypes and viral adaptations on HIV-1 infection outcome. Although there is limited data showing the impact of *ERAP2* on

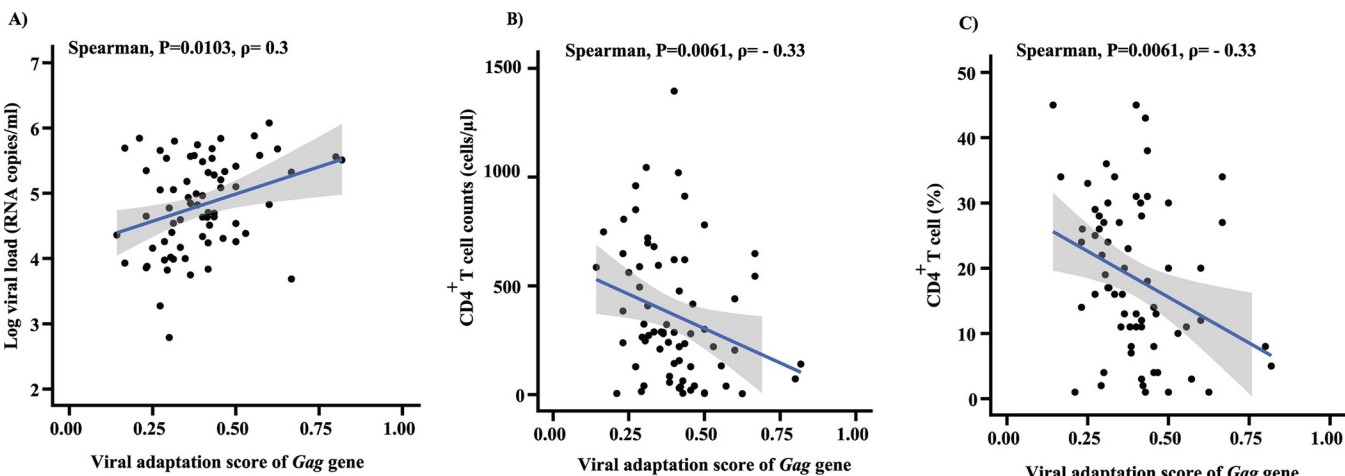

**Fig 3. HIV-1 adaptation to HLA-restricted immune response in *Gag* predicts disease severity. A)** Significant correlation of the proportion of viral adaptation in *Gag* with log viral load RNA copies/ml (P = 0.0103, ρ = 0.3; Spearman correlation test). **B)** Correlation of CD4+ T cell count with *Gag* adaptation score (P = 0.0061, ρ = -0.33; Spearman correlation test). **C)** Correlation of CD4+ T cell proportion with *Gag* adaptation score (P = 0.0061, ρ = -0.33; Spearman correlation test). The adaptation score ranges from 0 = non-adapted to 1 = fully adapted. Only sequences with ~100% coverage of protein sequence are included in this analysis (n = 70).

**Table 1. Cohort demographics.**

| Parameter | Description | Number (%) |
|---|---|---|
| Sex | Male | 193 (78.78) |
|  | Female | 52 (21.22) |
| Race | Caucasian | 173 (73.62) |
|  | ^Other | 62 (26.38) |
| CCR5D32 deletion | Present | 40 (16.33) |
| HLA-B*57, HLA-B*27 | Present | 39 (15.92) |
| HIV-1 subtype | B | 188 (76.73) |
|  | C | 21 (8.57) |
|  | AE | 20 (8.16) |
|  | A1 | 1 (0.41) |
|  | A | 2 (0.82) |
|  | G | 2 (0.82) |
|  | Other recombinants | 11 (4.49) |
|  |  | Median (IQR) |
| Log viral load (log copies/ml) |  | 4.69 (4.03–5.44) |
| CD4$^+$ T cell counts (cells/μl) |  | 360.5 (168.0–581.25) |
| CD4$^+$ T cell proportion |  | 19 (11–27) |

^Other includes African: n = 21, Indigenous Australian: n = 14, Asian: n = 25 and 16 participants with unknown race. IQR = Interquartile range.

antigen presentation, it has been shown that viral adaptations in the flanking region of CD8$^+$ T cell epitopes can prevent optimal peptide trimming by *ERAP1* [30]. Such changes to the repertoire of T cell epitopes presented by HLA molecules may impact the sites of HIV-1 adaptation. To investigate this relationship, we fitted an ANOVA model with an interaction term adjusted for sex, race, CCR5Δ32 and HIV-1 subtype to investigate the possible interaction effect of adaptation score with *ERAP2* haplotypes. Seventy participants with full sequence coverage of the known adaptation sites for specific HLA alleles [31] across *Gag* were used in this analysis. There was a significant interaction effect between variants of *ERAP2* and the level of the adaptation score in *Gag* on viral load (P = 0.0404, Fig 4A); this effect was most evident when applying a dominant model (haplotype AA vs carriers of haplotype B, P = 0.0111; Fig 4B). This interaction effect was lost when additional participants were added to the analysis with less than 90% sequence coverage of the adaptation sites in *Gag* (P = 0.4895) and may be due to missing data for key adaptation sites, as it has been shown that a single epitope in the *Gag* gene can reduce viral load by log$_{10}$ 0.21 [32]. There was no indication of a bias in the location of unsequenced regions of *Gag* (S2 Fig). Note, carriage of the *ERAP2* haplotype B was associated with lower *Gag* adaptation score (P = 0.0409, ANOVA), but this was driven by a small number of data points. There were no statistically significant interactions of *ERAP2* haplotypes and adaptation score in *Gag* on CD4$^+$ T cell count or proportion.

## *ERAP2* haplotypes associate with different sites under putative immune selection pressure

As an approach to indirectly assess if different *ERAP2* haplotype combinations alter peptide repertoire, we separated the participants into those that have the *ERAP2* AA haplotype (n = 68) and those that carry the B haplotype (n = 180) and then examined the *Gag* sequences in each group to identify HIV-1 polymorphisms associated with specific HLA alleles. Here, all

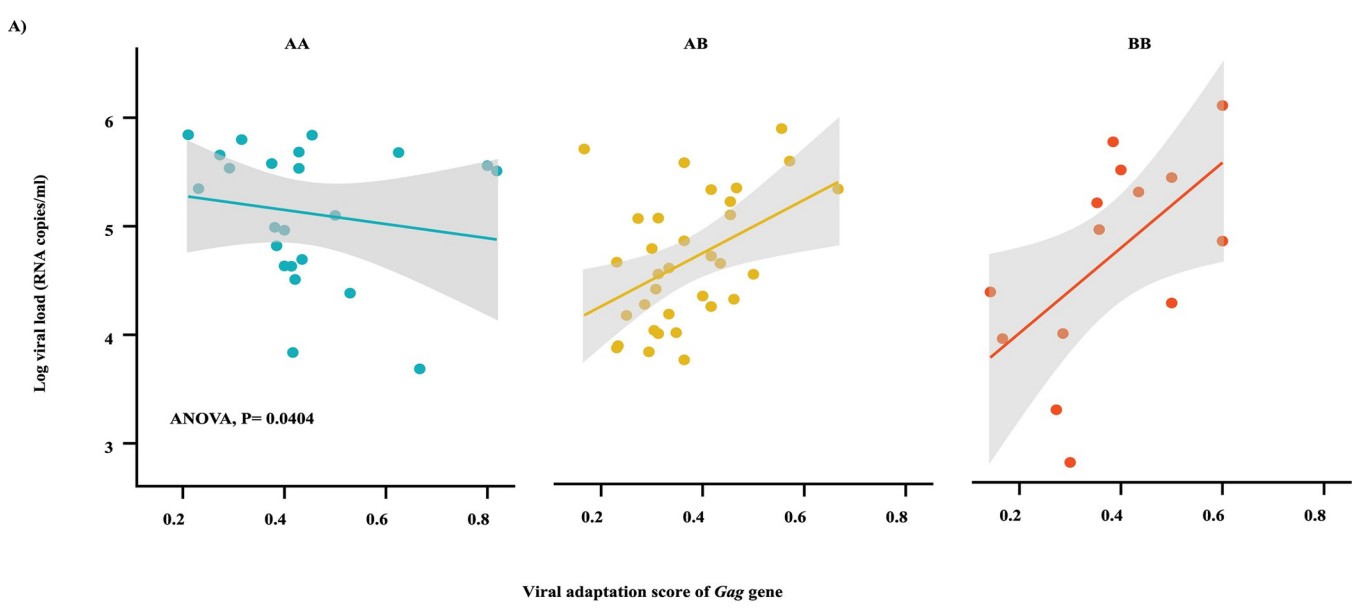

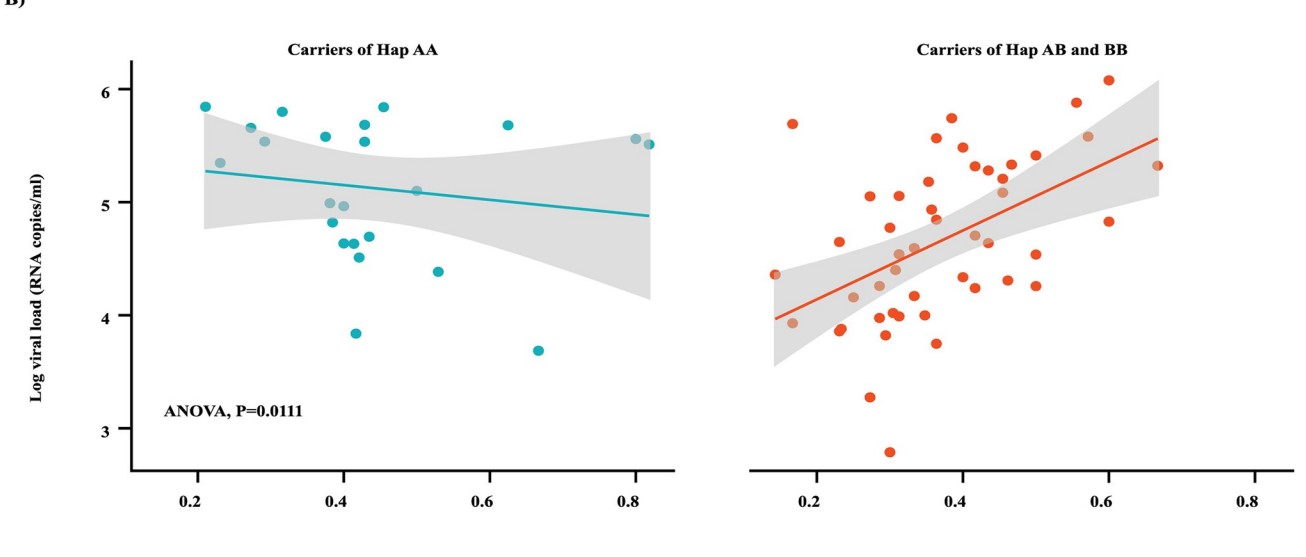

**Fig 4. Viral load is significantly affected by the interplay between *ERAP2* haplotypes and the level of *Gag* adaptation to HLA class I-restricted immune response. A)** Shows adaptation patterns in participants vary in their *ERAP2* haplotypes with *Gag* adaptation score can result in significantly different outcome using the codominant model. **B)** Distinct adaptation patterns are seen with carriers of haplotype B using dominant inheritance model. Analysis was performed by fitting ANOVA model with an interaction term adjusted for sex, race, CCR5Δ32 deletion, HIV-1 subtype for 70 participants with complete adaptation profile for *Gag*. Hap A = *ERAP2* haplotype A, Hap B = *ERAP2* haplotype B.

available *Gag* sequences were used as individual sites were examined. Polymorphisms at 66 positions in the *Gag* protein were associated with 102 HLA alleles in haplotype AA group and 58 positions were associated with 83 HLA alleles in the group that carry haplotype B (P < 0.05; fisher exact test). The associations for the two groups overlap in only six positions (positions 12, 26, 28, 242, 357). Correction for multiple comparisons was not applied here due to the small number of participants.

To investigate the likely relevance of the HLA associations found, we determined if the associated residue fell within experimentally defined epitopes restricted by the same HLA allele

(https://www.hiv.lanl.gov/content/immunology) or if the region is predicted to bind to the specific HLA allele using NetMHCpan 4.1 [33] (S6 Table and S3 Fig). For the *ERAP2* haplotype AA group, 17.6% of the HLA-associated viral polymorphisms fell within known T cell epitopes and 29.4% within a predicted T cell epitope with the correct HLA restriction. Similarly, for the participants carrying the *ERAP2* haplotype B, 22.9% of the HLA-associated viral polymorphism sites fell within known T cell epitopes and 32.5% within a predicted T cell epitope with the correct HLA restriction. For the five positions that were predicted to fall in regions with strong binding affinity to HLA, two sites were associated with a reduced binding affinity score with change from the consensus amino acid. It should be noted that some of the association sites (33.3% in AA group and 22.9% in AB/BB group) were found to be in the flanking region of epitopes (within 10 amino acid positions of the N- and C-terminal flanking regions), potentially affecting peptide processing [30, 34, 35]. The results suggest some overlap in the presented peptide repertoire by individuals with either *ERAP2* haplotype AA or carrying haplotype B, but also distinct peptide presentation by specific HLA and *ERAP2* allelic combinations.

## Pattern of T cell immune responses in hosts with different *ERAP2* haplotypes suggests overlapping immune targets

Another indirect measure we used to assess if carriage of *ERAP2* haplotypes affects the peptide repertoire was to examine *ex-vivo* HIV-1-specific T cell responses in a cohort with known HLA and ERAP genotypes. We re-examined the HIV-1-specific T cell responses in 15 individuals with HIV infection in our previous studies [36–38], in which those with the same HLA genotype were tested with a series of peptides representing known T cell epitopes across the HIV-1 proteome, including *Gag*. In these previous studies, antigen-specific recognition and/or responses were identified using specific HLA class I allele-peptide tetramers or IFNγ capture via intracellular cytokine staining. Of the 15 participants, two carried the haplotype AA combination with only one peptide eliciting an IFNγ response above background from eight different peptides (12.5%). For the five participants with haplotype BB there were four IFNγ responses to peptides from 13 different tests (30.77%). Interestingly, for participants with the *ERAP2* haplotype AB, we observed 14 positive IFNγ responses out of 20 tests (70%) [36] (S7 Table). These results suggest overlap in the T cell immune targets for participants with different *ERAP2* haplotype combinations.

## *ERAP2* variants and viral adaptation in *Gag* gene explain a substantial proportion of the variation observed in viral load

Previous studies have shown that variations at the CCR5 locus and HLA loci are significantly associated with viral load and explained a proportion of the variance observed in viral load ranging from 13% (increasing to 22% when including demographic factors) [2] to 14.5% (increases to 25% when including additive host genetic effects) [4]. In line with these findings, sex, CCR5Δ32 and carriage of HLA protective alleles (HLA-B*57 and -B*27) showed a significant association with differences in viral load in this cohort, explaining 17.67% of the observed variation (S8 Table). Adding *ERAP2* haplotypic variation increased the total explained variance by 5.2%. Furthermore, in the cohort subset (n = 70) with full viral sequence coverage at the adaptation sites, HIV-1 subtype, viral adaptation score in *Gag* and *ERAP2* haplotypes increased the explained proportion of the variance to 39.34%. Adding the interaction effect between *ERAP2* haplotypes and adaptation score in *Gag* into the model increased the explained variance in viral load to 45.1%, in keeping with the interaction between ERAP 2 and HLA being epistatic and mediated by modifying HLA-associated viral adaptation patterns.

### Examination of the impact of *ERAP2* SNPs alone and the interaction effect with viral adaptation on spVL in other HIV-1 cohorts

We sought to replicate these findings in independent datasets, where host genetic and linked clinical data were collected and available. We obtained data derived from multiple cohorts collected as part of 12 genome-wide association studies of HIV-1 outcome, eight combined under the International Collaboration for the Genomics of HIV (ICGH) as described in [39] (S10 Table). It should be noted that several of these cohorts were of non-European ancestry, where HLA and other genetic loci display significant variation in allele frequencies to the WAHIV cohort. Furthermore, these studies did not include HIV sequencing, so analyses of HIV adaptation were not possible. *ERAP2* rs2549782 SNP was genotyped in 14 out of the 21 cohorts only and there was no genotyping for the tag SNP of haplotype B, rs2248374. Using an ANOVA analysis adjusted for sex where applicable, no significant association was found between *ERAP2* rs2549782 genotypes and spVL in these cohorts (S10 Table).

In the remaining seven cohorts that did not include *ERAP2* rs2549782 genotyping, we used proxy SNPs of rs2248374. In six of those cohorts, we found proxy SNPs that had high LD with rs2248374 (S11 Table). After performing an ANOVA analysis adjusted for sex where applicable, no significant association was found between the proxy SNPs and spVL.

Data from the Swiss HIV Cohort Study (SHCS) had broadly matching ancestry, genome wide SNPs, HIV subtype B sequences and a derived value for spVL in 163 participants [40]. We performed HLA-class I imputation to 4 digit resolution using the "HIBAG" package in RStudio [41]. Utilising host HLA-class I genotypes and autologous *Gag* HIV-1 sequence for each participant in this cohort, we calculated HLA-associated HIV-1 adaptation score of *Gag* while also considering viral sequence coverage at the possible adaptation sites. There was no significant association between *Gag* adaptation score and spVL (P = 0.1477, ρ = -0.16; Spearman correlation test). We next fitted an ANOVA model including sex as a covariate to investigate the interaction effect between *ERAP2* rs2549782 genotypes and viral adaptation score of *Gag* on spVL. There was no significant interaction effect detected in this analysis (P = 0.6583; ANOVA).

### Discussion

Many studies have emphasized the dominant role of HLA on HIV-1 disease outcome [42] and the impact of HLA-restricted T cell immune pressure on the evolution of HIV-1 [18]. There are comparatively few studies on ERAP 1 and 2, though these enzymes are fundamental to the HLA genotype-specific immunopeptidome. In this study of an Australian population, where the effects of known HLA and CCR5 determinants are evident, we show that genetic variation in *ERAP2* additionally influences HIV-1-infection outcome. Furthermore, we showed a significant interaction effect on the impact of HIV *Gag* viral adaptation on pre-treatment viral load among 70 participants carrying different versions of the *ERAP2* gene. Research in other disease settings has revealed epistatic interactions between HLA and ERAP [14,43]. ERAPs can alter the peptide presented by a specific HLA allele or redirect the response to novel T cell epitopes [9]. With regards to HIV, a previous study showed that viral adaptations can affect epitope processing, including trimming by ERAPs in commonly targeted regions in *Gag* (P17 and P24), thereby affecting the immunodominance of CD8[+] T cell responses [44]. In addition, *ERAP1* and *ERAP2* haplotype A trimming activity has been found to be significantly affected by the internal sequence of the targeted peptide [14,45], and high viral adaptation will likely impair peptide trimming by ERAP.

The interaction effect observed in this study was significant in participants who carried *ERAP2* haplotype B (AB and BB haplotypes) showing a linear relationship such that the viral

load increases as the level of adaptation in the *Gag* gene increases, but this pattern was not observed for participants with *ERAP2* haplotype AA. As the adaptation score is calculated based on a list of known HLA-associated polymorphisms [31], it is plausible that subdominant or novel T cell epitopes are targeted in hosts who carry haplotype A. This is supported by experimental evidence showing that *ERAP2* haplotype A enzymatic activity is influenced by variation in the rs2549782 SNP that causes changes in the specificity and activity of the *ERAP2* enzyme [26]. Importantly, this change is equivalent to two orders of magnitude, resulting in an alteration in the peptide repertoire and impairment of the capacity of this enzyme to complement epitope processing mediated by *ERAP1* [26]. In another study that investigated the underlying cause of Birdshot Uveitis, an autoimmune disease associated with *ERAP2* haplotype A [46], the authors found that *ERAP2* haplotype A generates an HLA-A29-specific antigen repertoire by increasing the abundance of a distinct motif (with F and Y amino acids at P2) [14]. Our analysis of HLA-associated polymorphism in the *Gag* showed limited overlap in the associated amino acid positions between *ERAP2* haplotypes, suggesting *ERAP2* variation likely influences the repertoire of HLA-peptide presentation.

We also showed that *ERAP2* haplotype B, which encodes truncated *ERAP2* transcripts, predicts a lower HIV-1 viral load. Earlier studies have reported that the *ERAP2* protein encoded by haplotype B undergoes nonsense-mediated decay [25], but more recent work has shown that virally-infected cells (with influenza or HIV-1) can trigger distinctive short transcripts of *ERAP2* (termed *ERAP2*/Iso3 and *ERAP2*/Iso4) from haplotype B that could partially replenish the loss of *ERAP2* expression [17,47]. The RNA expression of *ERAP2*/Iso3 can be translated into protein and correlates with the abundance of viral antigens (P24 antigen for HIV-1 and overall gene segments for the flu virus) *in vitro* [17,47]. Although the biological effect of *ERAP2*/Iso3 is still unclear, *ERAP1* and *ERAP2* interact to form heterodimers that enhance peptide trimming efficacy [48,49] and a possible interaction of *ERAP2*/Iso3 with *ERAP2*/haplotype A or with *ERAP1* has been proposed (reviewed in [50]). Ye et al. [47] suggested that the translation of these short isoforms that lack the aminopeptidase domain might result in a "dominant negative" effect to increase the immunogenicity of the peptide repertoire in the case of viral infection.

The role of *ERAP2* in biological processes other than antigen presentation (reviewed in [50]) suggest other immunological pathways involved in HIV-1 pathogenesis may also be affected by variation in *ERAP2*. A recent report presented evidence that *ERAP2* haplotype A conferred protection during the Black Death [16], while haplotype B predicts better prognosis in HIV-1 infection and lowers risk of chronicity in Hepatitis C [51], suggesting that bacterial and viral pathogens may elicit distinct ERAP expression profiles, as shown for haplotype B (*ERAP2*/Iso3) expression [17,47].

The *ERAP2* haplotype A may exert effects relevant to HIV-1 infection but not for chronic T cell responses. Studies of HIV-1 exposed seronegative individuals have found that *ERAP2* genotype AA is overrepresented, suggesting possible effects operating at acquisition of infection and distinct from antigen processing for antiviral T cell responses [52,53]. In addition, *in-vitro* infection of peripheral blood mononuclear cells (PBMCs) with HIV-1 showed lower p24 levels in *ERAP2* AA than in AB and BB genotype backgrounds; however, relevant interactions with other factors such as ERAP/iso3 heterodimers and global T cell responses are not captured in such *in-vitro* studies [53].

Our study does confirm previous literature regarding the impact of viral adaptation on disease prognosis [19,20]. While there was no correlation between clinical measurements and the level of viral adaptation of *Pol* and *Nef* genes, there was a correlation signal detected for the *Gag* gene. This result supports previous findings that *Gag* is an important immunogen [54] and the breadth of *Gag*-specific CD8[+] T cell responses are associated with low levels of viremia

[32]. Furthermore, the presence of *Gag*-specific CD8$^+$ T cell responses ($>$ two T cell targets) delays onset to AIDS [32]. Targeting specific *Gag* epitopes likely imposes strong selection pressure on HIV-1 [32], and adaptation occurring in this region likely results in sufficient loss of immune control that would affect clinical measurements [19].

We did analyse other datasets derived from GWAS with comparable data, though in ethnically distinct populations. While GWAS have contributed largely to understand the role of HLA in controlling HIV-1 infection [2–4,40,55], there are limitations in detection of determinants with moderate effect size in reaching genome-wide significance, and with interactive effects such as epistasis (reviewed in [56]). Furthermore, the effects of genes subject to significant transcriptional regulation under certain biological conditions such as inflammation, may be underestimated in GWAS. The impact of this limitation in GWAS is apparent in studies examining Crohn's disease in which applying a genetic model that allows epistasis interactions could account for around 80% of heritability in this disease, whereas the many loci identified via GWAS explain only a minor percentage of disease heritability [1]. Of note, *ERAP2* SNPs (rs2549782 and rs2248374) have not been identified through previous GWAS examining the role of host genetic factors on HIV-1disease outcomes [2–4,40,55]. Our analysis with GWAS HIV-1 datasets reported in [39,40] did not show significance for *ERAP2* rs2549782 SNP associations with spVL. This may be attributed to differences in the definition of spVL. While in our study, the viral load is defined as the first pre-treatment viral load measurement taken at clinical presentation; in the replicated cohorts, spVL was defined as the mean of at minimum two viral load measurements taken during the chronic phase of infection [2,39,40,55] with some including the criteria of using VL data over a period of three years and diverging by no more than 0.5 log [2]. Another possible reason might be related to different combinations of *ERAP2* haplotypes in the other cohorts and WAHIV cohort. *ERAP2* is a highly polymorphic genetic region with two major haplotypes that branches further into other haplotypes with low frequencies [52,57]. A study conducted to investigate the lack of association of the rs2549782 SNP with preeclampsia in Chileans showed that, unlike other populations, there is a lack of LD between rs2549782 and rs2248374 SNPs, reflecting different *ERAP2* haplotypes [58]. Our analysis of the proxy SNPs (S11 Table) shows that although the two proxy SNPs rs2548533 and rs2549797 are in complete LD with rs2248374, there was a markedly different minor allele frequency in a cohort of European ancestry compared with others reported in the 1000 Genomes project [59], suggesting there may be different *ERAP2* haplotype structures in different populations. Furthermore, evidence of the distinctive influence of *ERAP2* based on HLA binding specificities in Ankylosing Spondylitis studies [60,61] suggests the likelihood of different HLA alleles and/or *ERAP1* and 2 haplotypes in our study and other cohorts limits the ability to replicate results with existing data. In addition, several of the GWAS were in populations of non-European ancestry in which the prevalence of HLA alleles and ERAP variants differ significantly and this may influence the list of relevant SNPs and viral adaptations used in the analyses.

The association between overall autologous HLA-associated HIV-1 adaptation score (calculated for near full-length of HIV-1 genome) and HIV-1 clinical outcome has been well documented [19]. The fact that we did not find a significant association between spVL and the adaptation score in the SHCS cohort is not unexpected. Carlson et al. [19] reported that *Gag*-specific adaptation score was significantly associated with VL (defined as a cross-sectional single measurement viral load taken mostly upon first presentation) in the Southern Africa (HIVC) cohort (P = 0.03) but not in the British Columbia (HIVB) cohort (P = 0.51), although in both cohorts the overall adaptation score was significantly associated with VL (P = 1*10$^{-16}$, Southern Africa (HIVC) cohort; P = 0.03, British Columbia (HIVB) cohort) [19]. Notwithstanding these multiple and significant limitations in replication analyses in other cohort

datasets, further population analyses and experimental studies are required to validate our findings and assess the generalizability in other chronic infections. We had insufficient statistical power to examine interaction between different combinations of specific HLA alleles, ERAPs and T cell epitopes, which may underpin global epistatic effects and account for population differences. Given the strong likelihood that epistasis and other interactions between genetic loci operate *in-vivo*, further studies are important not just for understanding HIV pathogenesis but also the broader ways in which antigen processing contributes to immune defenses against human pathogens.

## Materials and methods

### Study design and participants

Participants (n = 249) for this study are a subset of participants from the WAHIV cohort; a prospective observational study that commenced in 1992 [22]. The stratification criteria included participants who were treatment naïve, had well documented demographic information and pre-ART clinical data (viral load and CD4$^+$ T cell count and/or proportion) (Table 1). Plasma and peripheral blood mononuclear cells (PBMCs) were isolated from whole blood and stored at -80˚C. The participants in this study were predominantly male (78.78%), of Caucasian origin (73.62%) and infected with HIV-1 subtype B (76.73%). Race was self-reported.

### Ethics statement

Ethics approval for the conduct of this study was obtained from the Royal Perth Hospital Human Ethics Committee (EC2004/005) and Murdoch University Human Research Committee (2014/048; 2017/242). The protocol and the procedures of the study were conducted in conformity with the ethical guidelines of the World Medical Association Declaration of Helsinki. Participants enrolled in this study gave written informed consent prior to study commencement.

### DNA and viral RNA extraction

Genomic DNA was extracted from PBMCs using the DNA Isolation Kit (Beckman Coulter, Brea, CA, USA) and viral RNA was extracted from plasma samples using the MagMAX-96 viral RNA Isolation kit (Thermo Fisher Scientific, Waltham, MA, USA), according to the manufacturers' instructions.

### Viral sequencing

Viral RNA was converted to cDNA using the SuperScript III One-Step RT-PCR System with Platinum Taq High Fidelty Kit (Life Technologies, Carlsbad, CA, USA), as recommended in the manufacturer's instructions. Samples were then amplified using a nested PCR approach to target the HIV-1 genes *Pol*, *Gag* and *Nef* as previously described [20]. Samples were sequenced either via the sanger-based method or using the 2×300bp paired-end chemistry kit on the Illumina MiSeq sequencer (Illumina, San Diego, CA, USA). Raw sequencing reads were filtered and analysed using the visual genomics analysis studio (VGAS) tool [62]. HIV sequences have been submitted to GenBank and the relevant accession numbers are available in S2 Data.

### HLA genotyping

Genotyping of the HLA class I (HLA-A, -B and -C) and class II (HLA-DRB1) loci was based on exon 2 and 3 using locus-specific PCR amplification primers [63]. Where the HLA type was ambiguous, primers specific for HLA subtypes were used to perform allele-specific PCR

followed by sequencing to resolve typing ambiguity. Assign (Conexio Genomics, WA, Australia) software was used to interpret sequencing chromatograms.

### *ERAP* genotyping

Fourteen *ERAP1* (rs3734016, rs73148308, rs26653, rs27895, rs2287987, rs27434, rs73144471, rs27529, rs78649652, rs30187, rs10050860, rs111363347, rs17482078, rs375081137) and two *ERAP2* SNPs (rs2248374 and rs2549782) were genotyped using next generation sequencing as previously described [20]. An additional genotyping step was performed for the *ERAP1* SNP rs26618 using the predesigned rhAmp genotyping assay [64] (Integrated DNA Technologies, Inc., USA). Instruction of the rhAmp assay reaction mixture and PCR cycling conditions were according to the manufacture's protocol. Controls for the assay included genomic DNA extracted from cell lines (CIR, RML (IHW09016) and WT100BIS (IHW09006)) and an amplified genomic DNA sample (ACH-2) with confirmed rs26618 genotype combinations using next generation sequencing. Data were analyzed using the CFX manager 3.1 software (Bio-Rad Laboratories Pty Ltd).

### *ERAP1* and *ERAP2* haplotype construction

*ERAP1* and *ERAP2* haplotypes were constructed based on the genotyping of 10 non-synonymous variants using the Haplo.Stats package version 1.8.9 [65] in RStudio version 4.2.3 that computes maximum likelihood estimates of haplotype probabilities using the expectation maximization algorithm (EM) [66,67]. The estimated haplotypes and their frequencies were compared to previously reported haplotypes [13], with estimated haplotypes with frequencies lower than 0.5% excluded from the analysis (S2 and S4 Table). The estimated haplotypes were used to infer the haplotype combination for each participant in the cohort and the haplotype pair with the highest posterior probability was selected for subsequent analysis.

### HIV-1 subtype assignment

HIV-1 subtype was obtained from clinical records and confirmed using a combination of bioinformatic tools that infer subtype using different methodologies: REGA (phylogenetic-based tool) [68], COMET (statistical-based tool) [69], QC LANL (sequence quality check tool, https://www.hiv.lanl.gov/content/sequence/QC/index.html) and MEGA X (software for manual check of the possible subtype by constructing phylogenetic trees) [70]. HIV-1 subtype was assigned where there was concordance between the clinical records and the different bioinformatic tools.

### Calculation of viral adaptation score

HIV-1 adaptation scores were calculated as previously described [20] and were used to assess the level of HLA-associated HIV-1 adaptation. Adaptation was based on a previously identified list of variations within the HIV-1 genome that were statistically associated with carriage of specific HLA alleles at the population level [31]. Adaptation scores were calculated for each participant by dividing the number of adaptations present within their viral sequence for the *Gag*, *Pol* and *Nef* genes, separately, by the total number of possible adaptations based on an individual's HLA alleles. Only those sequences that included 100% of the potential adaptations for each HIV-1 gene were selected to perform the association analysis, as this removes any bias in the sequence coverage across a gene and provides equal coverage of putative adaptations for all participants. It should be noted that the adaptation list contains sites identified for HIV-1 subtype B (76.73% of infections in this cohort), but it is likely that there will be overlaps in T

cell targets between subtypes [21,71–73]. However, to account for a possible bias in the association analysis, viral subtype (coded as subtype B and non-B) was included as a covariate whenever the adaptation score was used in an analysis. Furthermore, results were confirmed by limiting the analysis to subtype B only when the sample size was sufficient to perform the analysis.

## Estimating linkage disequilibrium

To avoid the confounding effect of linkage disequilibrium (LD) in the association analysis, we first sought to identify the LD patterns in the host immune genetic data in the cohort. For HLA class I (HLA-A, -B, -C) and class II (HLA-DRB1), an online tool was used to explore LD patterns using fisher exact test comparisons (https://www.hiv.lanl.gov/content/immunology). For investigating LD among *ERAP1* and *ERAP2* SNPs, HaploView version 4.2 software [74] was used to compute D' and $r^2$ values for SNPs of interest. Fig 1 shows the results of the analysis. A cut-off D' value >0.80 was set to indicate LD.

## Hardy-Weinberg equilibrium test

Deviation from Hardy-Weinberg equilibrium (HWE) was examined for the HLA alleles and the *ERAP1* and *ERAP2* SNPs with an alpha value less than 0.05 considered statistically significant for HWE deviation. For the HLA data, the analysis was done in RStudio version 4.2.3 using the midasHLA package version 1.4.0 [75] that is specifically designed for HLA data. Testing the *ERAP1* and *ERAP2* SNPs for HWE deviation was performed using the SNPassoc package [76] based on a method that computes an exact test for HWE and controls for type I error regardless of sample size [77]. Deviation from HWE was detected in 13.6% of HLA class I and II alleles, which is expected as the HLA loci are under selection [78]. All the tested *ERAP1* and *ERAP2* SNPs were in HWE (S9 Table).

## Statistical analysis

### Normality

To approximate normal distribution some variables were transformed; viral load was transformed using $\log_{10}$ transformation and CD4$^+$ T cell counts were transformed using the Box-Cox method [79] with an estimated transformation parameter (lambda) equal to 0.4628. Following transformation, $\log_{10}$ viral load remained slightly left-skewed and CD4$^+$ T cell counts were slightly right-skewed. Therefore, a non-parametric test was applied where appropriate and model assumptions were checked for every model implemented using diagnostic plots, particularly for assumption of the normality of the residuals of the model.

**Covariates.** Heterogenicity in pre-treatment clinical markers of HIV-1 infection has been shown to be affected by sex and race [80] as well as a deletion in the CCR5 gene (CCR5Δ32) [2,4]. Therefore, to avoid cofounding effects on the analysis, the above-mentioned variables were included as covariates in models where disease outcome was used as a response variable in addition to controlling for the HIV-1 subtype whenever adaptation score was used in the model. Race factor was coded as Caucasians and 'other', as the majority of the cohort was of Caucasian origin (73.62%) and other racial groups were in low frequencies (26.38%). An alpha value <0.05 was considered significant. P values were corrected for multiple comparisons using the FDR approach. All data analyses were performed in RStudio, version 4.2.3

**Tests.** The independent association of *ERAP1* and *ERAP2* SNPs with disease outcome was done using ANOVA and adjusted as above with different genetic inheritance models in the SNPassoc package [76] in RStudio. Loci that were monomorphic or showed genotype

variability <1% were excluded from the analysis. Also, SNPs that had <2% of the minor allele frequency for any genetic inheritance model were excluded from the analysis.

*ERAP2* haplotype association with either disease outcome or viral adaptation score was done using ANOVA analysis and model assumption checked using diagnostic plots. Interaction effect of *ERAP2* haplotypes and viral adaptation score on disease outcome was done by fitting the ANOVA model with an interaction term.

Differences in the level of adaptation score across different HIV-1 genomic regions was examined using the Kruskal–Wallis test followed by Dunn's Test of multiple comparisons for post-hoc analysis. A Spearman's rank correlation coefficient test was used to examine the association of gene-specific adaptation score with disease outcome. Fisher exact test was done to examine the association of variation in the HIV *Gag* gene (consensus vs non-consensus sequence) with HLA among different *ERAP2* haplotypes at the 0.05 significance level.

The total variance explained by host and viral genetic factors (sex, race, carriage of CCR5Δ32, HLA-B*57 and B*27, carriage of different *ERAP2* haplotypes, adaptation score of *Gag* gene, HIV-1 subtype) was estimated by fitting multiple regression models and assessing the coefficient of determination ($R^2$) values. In addition, the change in adjusted $R^2$ values were compared for each model after adding additional covariates. S8 Table shows all the regression models used, the $R^2$ and the adjusted $R^2$ values.

## Supporting information

**S1 Table. Association of *ERAP1* and *ERAP2* SNPs with HIV-1 clinical markers**
(XLSX)

**S2 Table. Estimated *ERAP2* haplotype distribution.**
(XLSX)

**S3 Table. Association of *ERAP2* haplotypes with disease outcome in study participants using a codominant inheritance model.**
(XLSX)

**S4 Table. Estimated *ERAP1* haplotype distribution.**
(XLSX)

**S5 Table. Correlation of *Gag*, *Pol* and *Nef* autologous adaptation score with disease markers.**
(XLSX)

**S6 Table. Association of HLA alleles with *Gag* sequence among individuals carrying different *ERAP2* haplotypes.**
(XLSX)

**S7 Table. IFN-γ responses to immunodominant HIV-1 T cell epitopes among individuals with different *ERAP2* haplotypes.**
(XLSX)

**S8 Table. Proportion of variance explained by host and viral genetic factors.**
(XLSX)

**S9 Table. Description of *ERAP1* and *ERAP2* SNPs and HWE outcomes.**
(XLSX)

**S10 Table. Description of HIV-1 GWAS dataset reported by McLaren et al. [39].**
(XLSX)

**S11 Table. *ERAP2* Proxy SNPs with near complete LD with rs2248374.**
(XLSX)

**S1 Fig. No significant difference in the level of viral adaptation to HLA-restricted immune response across HIV genome.** The adaptation list used in this calculation is based on statistical association of specific adaptation sites to HLA-restricted immune response described by [31]. Viral sequences with complete coverage for the adaptation sites were used in this analysis. Kruskal-Wallis test was performed to examine the difference between viral genes and the level of the adaptations to HLA class I-restricted immune responses.
(TIF)

**S2 Fig. Lack of bias in sequence coverage of the *Gag* region.** This map represents sequence coverage for 70 individuals (highlighted with the green side ribbon) with complete coverage of the possible adaptation sites, and additional 31 (highlighted with the yellow side ribbon) that have between 90 to 96% of sequence coverage at the adaptation sites. Missing sequences in the map colored in red, whereas blue represents the presence of sequence.
(TIF)

**S3 Fig. The distribution of the top significant associations of HLA alleles with *Gag* sequence among individuals carrying different *ERAP2* haplotypes derived from S6 Table.** A threshold of <0.01 p-value was applied to filter the top significant associations in both groups. EV = experimentally verified. NP = not predicted. WB = weak binder. N-terminal and C-terminal indicates sites within 10 amino acids flanking epitopes.
(TIF)

**S1 Data. Individual host genetic data.**
(XLSX)

**S2 Data. GenBank accession numbers.**
(XLSX)

## Acknowledgments

We thank Professor Paul J. McLaren at University of Manitoba for providing an ICGH dataset for analysis. We thank the authors colleagues at the Vanderbilt University Medical Center, TN, USA and the Institute for Immunology and Infectious Diseases, Murdoch University, WA, Australia.

## Author Contributions

**Conceptualization:** Marwah Al-kaabi, Rebecca Pavlos, Elizabeth Phillips, Simon Mallal, Mina John, Silvana Gaudieri.

**Data curation:** Marwah Al-kaabi, Pooja Deshpande, Rebecca Pavlos, Abha Chopra, Hamed Basiri.

**Formal analysis:** Marwah Al-kaabi, Silvana Gaudieri.

**Funding acquisition:** Mina John, Silvana Gaudieri.

**Investigation:** Marwah Al-kaabi, Pooja Deshpande, Rebecca Pavlos, Hamed Basiri, Jennifer Currenti, Eric Alves.

**Methodology:** Marwah Al-kaabi, Martin Firth, Silvana Gaudieri.

**Project administration:** Marwah Al-kaabi, Silvana Gaudieri.

**Resources:** Spyros Kalams, Jacques Fellay, Simon Mallal, Mina John, Silvana Gaudieri.

**Supervision:** Mina John, Silvana Gaudieri.

**Validation:** Martin Firth.

**Visualization:** Marwah Al-kaabi.

**Writing – original draft:** Marwah Al-kaabi, Silvana Gaudieri.

**Writing – review & editing:** Marwah Al-kaabi, Pooja Deshpande, Martin Firth, Rebecca Pavlos, Abha Chopra, Hamed Basiri, Jennifer Currenti, Eric Alves, Spyros Kalams, Jacques Fellay, Elizabeth Phillips, Simon Mallal, Mina John, Silvana Gaudieri.

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
