## [Decision Letter · Decision Letter 0]

9 Nov 2023

Dear Al-kaabi,

Thank you very much for submitting your manuscript "Epistatic interaction between ERAP2 and HLA modulates viral adaptation and HIV-1 disease outcome" for consideration at PLOS Pathogens. As with all papers reviewed by the journal, your manuscript was reviewed by members of the editorial board and by several independent reviewers. In light of the reviews (below this email), we would like to invite the resubmission of a significantly-revised version that takes into account the reviewers' comments.

Please pay particular attention to the concerns about small sample size and validating your findings in other datasets.

We cannot make any decision about publication until we have seen the revised manuscript and your response to the reviewers' comments. Your revised manuscript is also likely to be sent to reviewers for further evaluation.

Sincerely,

Daniel C. Douek

Academic Editor

PLOS Pathogens

Richard Koup

Section Editor

PLOS Pathogens

Kasturi Haldar

Editor-in-Chief

PLOS Pathogens

orcid.org/0000-0001-5065-158X

Michael Malim

Editor-in-Chief

PLOS Pathogens

orcid.org/0000-0002-7699-2064

Reviewer's Responses to Questions

**Part I - Summary**

Reviewer #1: Siegel et al. presents in their manuscript an independent association of host genetic variants in ERAP1-2 and HLA-associated viral adaptations with HIV outcomes (VL and CD4 counts). They also show an epistatic interaction between the two primary variables and the outcome (P=0.01). ERAPs are interesting loci as highlighted by other papers cited by the authors and have effects on infectious and autoimmune diseases (Strange et al. 2010; Klunck et al. 2022) It is therefore not surprising that it could be associated with HIV-1 outcomes. Though many studies have addressed the role of HLA on disease outcome, there are few studies on ERAP 1 and 2. It addresses how genetic variations in ERAP 2 likely modulate HIV-1 infection outcome beyond HLA effects. The study also confirms previous findings on level of viral adaptation to disease prognosis and correlation between clinical measurements and level of viral adaptations for Gag gene, supporting previous findings on the importance of Gag as an immunogen. The paper is well written and lucid.

Reviewer #2: Summary: The authors describe a host/viral genetic study in 249 PLWH from Western Australia. In an association analysis, they observed a significant association between two highly correlated exonic SNPs in ERAP2 (rs2549782 and rs2248374) and viral load. This was substantiated in a haplotype analysis comparing individuals carrying both effect alleles to those carrying other combinations. In a paired analysis of viral sequence and host HLA type, the authors observed an association between escape mutations in Gag and clinical markers. Using these data, the authors build an adaptation score for Gag and show that it significantly interacts with ERAP2 haplotype when influencing viral load. By analyzing combinations of HLA alleles, ERAP2 haplotypes and viral sequence data, the authors conclude that different combinations of HLA type and ERAP2 type results in distinct peptide presentation properties. Overall this is an interesting study that potentially provides novel insight into HIV control and evolution. However, I do have some questions and suggestions to enhance the clarity of the manuscript.

**Part II – Major Issues: Key Experiments Required for Acceptance**

Reviewer #1: Although the premise of this study is interesting, this reviewer’s concern is that they are based on a relatively small sample size and not validated in additional studies, thereby contributing to the large literature on HIV host genetics that remain a single finding without additional validation. Can the authors address this concern and validate in additional samples? There are many previously published studies with genome-wide data, HLA types and viral sequence data which can be mined for replication (eg. Brumme Lab, Carrington Lab etc). Small sample size and genetic results without additional validation is a major limitation of this study.

Reviewer #2: 1. One concern I have is the reproducibility of the ERAP2 association. The authors cite several GWAS of HIV viral load (refs 2-4) that have larger sample sizes that the study presented here. However, to my knowledge none of those identified ERAP2 SNPs as being associated with viral load. If those data are publicly available, it would be interesting to know if the ERAP2 SNPs are even nominally associated in those studies and show similar effect directions. This would increase the confidence in the reported associations and could be done without much additional effort.

2. I also wonder how robust the association is to potential bias due to population stratification. The authors do state that they account for race, however, they use a binary covariate (Caucasian vs Other) to achieve this. This may not be sufficient if the other category is itself quite diverse with varying viral loads in different ancestry groups. Indeed the frequency of the ERAP2 polymorphisms can vary up to 10% across continental ancestry groups (at least in 1000Genomes data) and the haplotype frequencies presented in table S2 are quite different. One way to assess this may be to compare mean viral loads between the ancestry groups and/or to perform a subset analysis in the Caucasian only individuals.

3. The statistical tests and thresholds used vary throughout the manuscript without much justification as to why the tests were chosen and when/ why it is appropriate to not account for multiple comparisons. For example, when comparing HLA and viral sequence polymorphism stratified by ERAP2 haplotype, I don’t understand why the Fisher’s exact test was used and no multiple comparison correction was applied. The authors suggest the sample size is limited but n=68 and n=180 are roughly inline with the full cohort size. It would be interesting to know if the strongest associations (i.e. those most likely to be true) are more often found in canonical or predicted epitopes.

4. The methods used to determine the amount of variability explained by the various model parameters is also not clear. In particular, when including the interaction term into the model they observe a dramatic increase in the variance explained (39%). However, it is not clear to me that this accounts for the fact that adaptation score is highly related to HLA type and therefore is independent of a large HLA effect. Some additional description of how this analysis was performed would be informative here.

**Part III – Minor Issues: Editorial and Data Presentation Modifications**

Reviewer #1: Minor comments

Table -1 – Important parameters and the N and % are listed. Column 3 header should be changed (Number (%)) may not be the correct heading for Log viral load and CD4+ T cell count etc.

Change-Results line 130-( Fig-1 (A?)

Change -Results line 155- (Fig -1 (B?)

Results line # 156 to line #172 describing “ERAP2 haplotype B carrying a truncated version of ERAP2 predicts better HIV infection outcomes”-are described in Figures 3A, 3B and 3C.

While Results line # 178 to Line #188 describing “viral adaptation to HLA restricted T cell immune response to Gag correlate significantly with clinical outcome “, are described in Figure 2. Suggest that Figures are numbered in the order as they are mentioned in the text.

Reviewer #2: Line 126 the use of the word ‘independent’ caused some confusion since the SNPs are highly correlated and therefore non-independent

It feels counter-intuitive to have Figure 3 precede figure 2 in the results

Table S2 lists amino amino acid alleles for rs2549782 where the text uses primarily nucleic acids. Using similar coding throughout would improve ease of interpretation

PLOS authors have the option to publish the peer review history of their article (what does this mean?). If published, this will include your full peer review and any attached files.

Reviewer #1: No

Reviewer #2: No
---

## [Decision Letter · Decision Letter 1]

4 Feb 2024

Dear Al-kaabi,

Thank you very much for submitting your manuscript "Epistatic interaction between ERAP2 and HLA modulates viral adaptation and HIV-1 disease outcome" for consideration at PLOS Pathogens. As with all papers reviewed by the journal, your manuscript was reviewed by members of the editorial board and by several independent reviewers. The reviewers appreciated the attention to an important topic. Based on the reviews, we may accept this manuscript for publication, providing that you modify the manuscript according to the review recommendations.

Reviewer 1 remains concerned validation should be performed.

The reviewer provides the details of a dataset you may access for this purpose:

"A recent paper by McLaren, P.J. et al. Nature 620, 1025–1030 (2023), cites access to individual-level genotyping data which is obtainable on request (jacques.fellay@epfl.ch)"

Sincerely,

Daniel C. Douek

Academic Editor

PLOS Pathogens

Richard Koup

Section Editor

PLOS Pathogens

Michael Malim

Editor-in-Chief

PLOS Pathogens

orcid.org/0000-0002-7699-2064

Reviewer Comments (if any, and for reference):

Reviewer's Responses to Questions

**Part I - Summary**

Reviewer #1: (No Response)

Reviewer #2: (No Response)

**Part II – Major Issues: Key Experiments Required for Acceptance**

Reviewer #1: Major comments

1. My original concern for this paper remains that the findings of this study must be validated in additional cohorts for it to contribute to literature on host genetic effects with HIV disease outcomes. The authors could directly contact the last authors with the original raw data to validate their results:

A recent paper by McLaren, P.J. et al. Nature 620, 1025–1030 (2023), cites access to individual-level genotyping data which is obtainable on request (jacques.fellay@epfl.ch).

• Other potential cohorts of European ancestry are listed in van Manen et al.: (Genome-wide association studies on HIV susceptibility, pathogenesis, and pharmacogenomics. Retrovirology 2012 9:70)

2. In Line 62 of the manuscript- Author summary, the word “subject” should be changed to “host” or “participant “according to the NIAID HIV language guide

Reviewer #2: (No Response)

**Part III – Minor Issues: Editorial and Data Presentation Modifications**

Reviewer #1: Authors should provide a data availability statement

Reviewer #2: The authors have sufficiently addressed my comments. I would be in favour of including the figures/tables provided in the response as supplementary materials.

PLOS authors have the option to publish the peer review history of their article (what does this mean?). If published, this will include your full peer review and any attached files.

Reviewer #1: No

Reviewer #2: No

Figure Files:

Data Requirements:

Reproducibility:

References:

---

## [Decision Letter · Decision Letter 2]

3 Jun 2024

Dear Al-kaabi,

Thank you very much for submitting your manuscript "Epistatic interaction between ERAP2 and HLA modulates viral adaptation and HIV-1 disease outcome" for consideration at PLOS Pathogens. As with all papers reviewed by the journal, your manuscript was reviewed by members of the editorial board and by several independent reviewers. In light of the reviews of Reviewer #1 we would consider the resubmission of a significantly-revised version that takes into account this reviewers' comments.

The Reviewer remains concerned that the original ERAP2 findings have not been validated. Thus, in order for this manuscript to be considered further, the manuscript title, abstract and discussion will have to be modified to clearly and unambiguously reflect that this association was not validated in 10 additional cohorts. We cannot make any decision about publication until we have seen the revised manuscript and your response to the reviewers' comments. Your revised manuscript is also likely to be sent to reviewers for further evaluation.

Sincerely,

Daniel C. Douek

Academic Editor

PLOS Pathogens

Richard Koup

Section Editor

PLOS Pathogens

Michael Malim

Editor-in-Chief

PLOS Pathogens

orcid.org/0000-0002-7699-2064

Reviewer's Responses to Questions

**Part I - Summary**

Reviewer #1: 1. The authors have reached out to new authors to include analyses on previously published GWAS data with ERAP2 SNPS. The ERAP2 findings unfortunately did not validate in additional cohorts. Based on their additional analyses their original findings are not substantiated even with the limitations mentioned. The authors should tone down the title, abstract and the rest of the paper substantially to reflect these updates. Include clearly in the abstract that these findings were only seen in one cohort and not validated further.

2. The word “subject” continues to be used in the supplementary data etc.

**Part II – Major Issues: Key Experiments Required for Acceptance**

Reviewer #1: (No Response)

**Part III – Minor Issues: Editorial and Data Presentation Modifications**

Reviewer #1: (No Response)

PLOS authors have the option to publish the peer review history of their article (what does this mean?). If published, this will include your full peer review and any attached files.

Reviewer #1: No
---

## [Editor Report · Decision Letter 3]

19 Jun 2024

Dear Al-kaabi,

We are pleased to inform you that your manuscript 'Epistatic interaction between ERAP2 and HLA modulates HIV-1 adaptation and disease outcome in an Australian population.' has been provisionally accepted for publication in PLOS Pathogens.

Best regards,

Daniel C. Douek

Academic Editor

PLOS Pathogens

Richard Koup

Section Editor

PLOS Pathogens

Michael Malim

Editor-in-Chief

PLOS Pathogens

orcid.org/0000-0002-7699-2064
---

## [Editor Report · Acceptance letter]

3 Jul 2024

Dear Al-kaabi,

We are delighted to inform you that your manuscript, "Epistatic interaction between ERAP2 and HLA modulates HIV-1 adaptation and disease outcome in an Australian population.," has been formally accepted for publication in PLOS Pathogens.

Best regards,

Michael Malim

Editor-in-Chief

PLOS Pathogens

orcid.org/0000-0002-7699-2064